# MULTI-TREATMENT EFFECT ESTIMATION WITH PROXY: CONTRASTIVE LEARNING AND RANK WEIGHTING

## ABSTRACT

We study the treatment effect estimation problem for continuous and multi-dimensional treatments, in the setting with unobserved confounders, but high-dimension proxy variables for unobserved confounders are available. Existing methods either directly adjust the relationship between observed covariates and treatments or recover the hidden confounders by probabilistic models. However, they either rely on a correctly specified treatment assignment model or require strong prior of the unobserved confounder distribution. To relax these requirements, we propose a Contrastive regularizer (Cr) to learn the proxy representation that contains all the relevant information in unobserved confounders. Based on the Cr, we propose a novel Rank weighting method (Rw) to de-bias the treatment assignment. Combining Cr and Rw, we propose a neural network framework named CRNet to estimate the effects of multiple continuous treatments under unobserved confounders, evaluated by the Average Dose-Response Function. Empirically, we demonstrate that CRNet achieves state-of-the-art performance on both synthetic and semi-synthetic datasets.

## 1 INTRODUCTION

Causal inference is widely applied for explanatory analysis and decision making, e.g., Precision Medicine (Raita et al., 2021), Advertisement (Lada et al., 2019), Education (Johansson et al., 2016) and Digital Economy (Nazarov, 2020). With accessible observation data, many existing algorithms accurately estimate the effect of binary treatment by adjusting the confounders (i.e., the common causes of treatments and outcomes) which rely on unconfoundedness assumption that all confounders are observed.

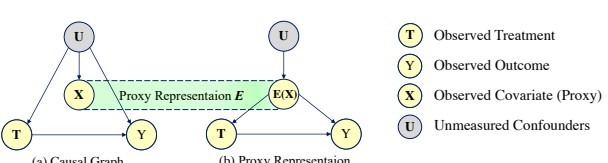

(a) Causal Graph     (b) Proxy Representaion

Figure 1: (a) Causal Structure of Raw Data, i.e., $Y \perp \mathbf{T} \mid \mathbf{U}$; (b) Target Relationship from *proxy representation*, i.e., $Y(\mathbf{T}) \perp \mathbf{U} \mid E(\mathbf{X})$.

However, continuous and multi-dimensional treatments and unmeasured confounders are common in practice. For instance, practitioners seek to develop precise medicine by studying the response of multiple drug dosages (i.e., treatment) on patient health state (i.e., outcome) (Shi et al., 2020). Besides, due to technique and manipulation issues, some key variables, associated with the treatments and outcomes, like patient's *immunity* maybe missing in the historical data, which are referred to as unmeasured confounders. To detect and adjust unmeasured confounders, practitioners would record some proxy variables (noised unobserved confounders, e.g., antibodies) which don't have a direct effect on treatments and outcome of interest but has a spurious association through shared common confounders (Fig. 1(a)).

In continuous treatments setting, under unconfoundedness assumption, recent works discretize the continuous treatment into multi-valued treatment (Hill, 2011; Wager & Athey, 2018) to traditional models, or develop generalize balancing methods for continuous scenario (Hirano & Imbens, 2004; Vegetabile et al., 2021; Huling et al., 2021). Among them, state-of-the-art works (Wu & Fukumizu, 2021; Schwab et al., 2020; Nie et al., 2021) learn a low-dimensional representation for raw data and

balance it using minimizing mutual information, which discard the imbalance part of raw data and lose most information for predictive task in practice. In fact, the technique implements a trade-off decreasing the estimator variance at the price of increasing the bias. Furthermore, with unobserved confounders, if we control the proxy rather than unobserved variables, the effect estimation will induce additional bias, referred as *recovery bias*. To deal with this bias, instead of balancing representations and discarding information to block the relationship between observed covariates and treatments, we propose a novel Contrastive regularizer (Cr) to learn a *proxy representation* for capturing all the relevant information in unobserved confounders with contrastive learning (He et al., 2020; Chen et al., 2020; Grill et al., 2020) which regularize representation space by *positive and negative pairs*. In Cr, we define the positive pair is the pair of treatments and proxies from the same sample, and the negative pair is the pair of treatment from one sample and proxies from different samples. And with an ideally representation for confounders, we would adopt a balancing methods to eliminate confounding bias, such as generalized propensity score (Hirano & Imbens, 2004).

However, one limitation is that the covariate balancing methods rely on the correct specified models. If we don't have any prior for the models of propensity score, i.e., the conditional distribution of treatment conditioning on the covariates, the effect estimation would still be biased, especially for high-dimensional data and continuous treatment. Besides, balancing methods still suffer from extreme values problem. Although recent methods (Fong et al., 2018; Vegetabile et al., 2021; Huling et al., 2021) propose to clip the score value or optimize balancing weights directly, they still fail in complex data, especially, under multi-continuous treatment setting. So a balancing method that have no extreme values and adapted to unobserved confounders is urgently needed. Therefore, to control for bias from treatment assignment, we propose to rank the weights obtained from inverse propensity score for more effective balancing weighting. Based on the proxy representation learned above, we sort the propensity score based weights in descending order and record their $rank$ (the order in sorted data) as rank weights (Rw), which is an effective and robust weights for treatment effect estimation, theoretically.

Combining Contrastive regularizer (Cr) and Rank weighting (Rw) methods, we propose a neural network framework CRNet to alleviate the outcome approximate bias in estimating the Average Dose-Response Function (ADRF). CRNet can accurately estimate the effects of multiple continuous treatments with high-dimension proxy variables. Empirically, we demonstrate that CRNet achieves state-of-the-art performance on both synthetic and semi-synthetic datasets.

## 2 RELATED WORK

**Causal effect identification with proxy methods**   Proxy (Guo et al., 2020) assumes that the unobserved confounders can be recovered from the observed covariates. CEVAE (Louizos et al., 2017), intact-VAE (Wu & Fukumizu, 2021) recover unobserved confounders with VAE (Kingma et al., 2019) constraint. Negative controls (Lipsitch et al., 2010) assume that there exist two negative control variables: one is related to treatments and confounders, and another is related to outcomes and confounders. DFPV (Xu et al., 2021) introduces neural networks to model the bridge function (Miao et al., 2018) for estimating the causal effect. The setting of this paper is similar to the proxy. But our method need no data distribution prior and outperforms others in performance.

**Estimation methods for continuous treatments**   For estimating the continuous treatment effect, a branch of methods include spline (Imai & Van Dyk, 2004), kernel methods (Flores et al., 2012), ensemble methods (Hill, 2011; Wager & Athey, 2018), representation-based methods (Schwab et al., 2020; Nie et al., 2021; Bica et al., 2020) model the relationship between treatments and outcomes. There is also a branch of methods (Hirano & Imbens, 2004; Imai & Van Dyk, 2004; Robins et al., 2000; Vegetabile et al., 2021; Arbour et al., 2021; Huling et al., 2021) aim at balancing the covariates shifts. Few previous works take into account of unobserved variables with continuous treatment assignment bias. In this paper, we propose the contrastive regularizer to gain the balancing methods with the presence of unobserved confounders. Also, we propose a new rank weighting method which have no extreme values and is not much sensitive to model misspecified. Combining Cr and Rw, we design a framework CRNet to estimate continuous treatment with proxy.

## 3 PRELIMINARIES

**Notation** For self-consistency, we use uppercase for random variables (e.g., $A$) and lowercase for their realization (e.g., $a$). We suggest bold the character $\mathbf{A}$ as a vector, otherwise a scalar. Given a variable $A_i^p$, superscript $^p$ represents the dimension of $A$, and subscript $_i$ denotes the i-th sample of $A$. $D_A$ refers to the total number of dimensions of $A$. $N_A$ means the total number of samples of $A$. Besides, a real valued sample $\{\mathbf{X}_i^p, \mathbf{T}_i^q, \mathbf{U}_i^r, Y_i\} \in \mathbb{R}^{p+q+r+1}$ denotes a random sample with observed covariates $\mathbf{X}_i \in \mathbb{R}^p$, treatments $\mathbf{T}_i \in \mathbb{R}^q$, unobserved confounders $\mathbf{U}_i \in \mathbb{R}^r$ and outcome $Y_i \in \mathbb{R}$. And $\{\mathbf{X}_i, \mathbf{T}_i, \mathbf{U}_i, Y_i\}_{i=1}^n$ refers to a set of $\{\mathbf{X}_i, \mathbf{T}_i, \mathbf{U}_i, Y_i\}$ with $n$ samples. $\mathbb{E}$ is denoted as expectation and $\mathbb{P}$ represents density distribution function. A calligraphic letter $\mathcal{H}$ is denoted as a hypothesis space.

### 3.1 PROBLEM SETUP

As Fig. 1(a) shown, in this paper, we focus on the treatment effect estimation problem for Continuous and multi-dimensional Treatments, in the setting with unobserved confounders, but high-dimension Proxy variables for unobserved confounders are available (Briefly, CTP problem). Specifically, we formalize proxy as:

**Definition 1** (Proxy) *The observed covariate and the noise view of unobserved confounder. Formally, $\mathbf{X} = f^*(\mathbf{U}, \epsilon_1)$, where the noise item $\epsilon_1 \perp \{\mathbf{T}, \mathbf{U}\}$ and $f^*$ means the true function.*

Without loss of generality, we focus on estimating the Average Dose-Response Function (ADRF) curve in this paper. We denote $ADRF^*$ as the true ADRF and define that

**Definition 2** (ADRF) *The potential outcome of continuous treatments over the population:*
$$ADRF^* = \mathbb{E}[Y_i(\mathbf{T}_i = \mathbf{t})] = \mathbb{E}[\phi^*(\mathbf{U}, do(\mathbf{t}))], \tag{1}$$
*where $do(\mathbf{t})$ means the do operation on treatment that $do(\mathbf{t}) \perp \{\mathbf{U}\}$.*

### 3.2 MOTIVATION

To analyse the complex CTP problem, we simplify the ADRF estimation considering an additive regression model $\phi(\mathbf{X}_i|\mathbf{t})$ given the observed $\mathbf{t}$ with no sample selection bias following Imai et al. (2008):
$$\phi(\mathbf{X}_i|\mathbf{t}) = \hat{\phi}(\mathbf{U}_i|\mathbf{t}) + \hat{h}(\epsilon_{1i}|\mathbf{t}) \tag{2}$$

And the the estimated $\hat{ADRF}$ of $\phi(\mathbf{X}_i|\mathbf{t})$ can be formulated as
$$\hat{ADRF} = \mathbb{E}[\phi(\mathbf{X}_i|\mathbf{t})], \tag{3}$$
where $\mathbf{t}$ means the observed treatment that $\mathbf{t} \not\perp \{\mathbf{U}\}$.

We set $\hat{ADRF}$ as baseline and define the estimation error as
$$\Delta = \mathbb{E}[\phi^*(\mathbf{U}, do(\mathbf{t})) - \phi^*(\mathbf{U}|\mathbf{t})] + \mathbb{E}[\phi^*(\mathbf{U}|\mathbf{t}) - \hat{\phi}(\mathbf{U}_i|\mathbf{t})] - \mathbb{E}[\hat{h}(\epsilon_{1i}|\mathbf{t})] \tag{4}$$
Given the Eq.(4) (the detailed derivation process is in the Appendix), we denote the first error term $\Delta_T = \mathbb{E}[\phi^*(\mathbf{U}, do(\mathbf{t})) - \phi^*(\mathbf{U}|\mathbf{t})]$ as the bias from treatment assignment, the second term $\Delta_Y = \mathbb{E}[\phi^*(\mathbf{U}|\mathbf{t}) - \hat{\phi}(\mathbf{U}_i|\mathbf{t})]$ as the bias from outcome approximate and the third term $\Delta_{\epsilon_1} = -\mathbb{E}[\hat{h}(\epsilon_{1i}|\mathbf{t})]$ as the bias from the recovery of $\mathbf{U}$. Thus, we decompose the estimation error $\Delta$ into
$$\Delta = \Delta_{\epsilon_1} + \Delta_T + \Delta_Y \tag{5}$$

Then we the divide ADRF estimation with multi-continuous treatments and proxy problem into three component: 1. Reduce the bias of recovery error $\Delta_{\epsilon_1}$ on ADRF estimation. 2. Reduce the bias $\Delta_T$ from treatment assignment of $\mathbf{T}$ on $\mathbf{U}$. 3. Reduce the bias $\Delta_Y$ from the outcome approximation.

### 3.3 ASSUMPTIONS

Throughout this paper, we assume the two common assumptions **Assumption 1** *Stable Unit Treatment Value Assumption, SUTVA* and **Assumption 2** *Overlap/Positivity assumption* (Imbens & Rubin, 2015) are satisfied. Moreover, we assume the following assumptions.

**Assumption 3** (Latent unconfoundedness) *The potential outcome is independent of treatment assignment given the unobserved confounders. Formally, $Y(\mathbf{t}) \perp \mathbf{T}|\mathbf{U}$.*

**Assumption 4** (Proxy assumption) *The proxy is independent of treatment and outcome given unobserved confoudner. Formally, $\mathbf{X} \perp \{\mathbf{T}, Y\}|\mathbf{U}$.*

To eliminate the recovery error $\Delta_{\epsilon_1}$, following Louizos et al. (2017), we consider it as a self-supervised representation learning problem: Recovering latent representation $\mathbf{U}$ from $\mathbb{P}(\mathbf{t}, \mathbf{X}, y)$, which means estimating $\mathbb{P}(\mathbf{U}|\mathbf{t}, \mathbf{X}, y)$, and we formulate this problem as $\mathbb{P}(y|\mathbf{t}, \mathbf{X}) = \int_{\mathbf{U}} \mathbb{P}(y|\mathbf{t}, \mathbf{X}, \mathbf{U})\mathbb{P}(\mathbf{U}|\mathbf{X}, \mathbf{t})d\mathbf{U} = \int_{\mathbf{U}} \mathbb{P}(y|\mathbf{t}, \mathbf{U})\mathbb{P}(\mathbf{U}|\mathbf{X}, \mathbf{t})d\mathbf{U}$. (For the discussion of proxy identification, see Appendix). Then we make assumptions that

**Assumption 5** (Recoverability) *The density $\mathbb{P}(\mathbf{U}, \mathbf{t}, y)$ of the latent confounders $\mathbf{U}$ can be approximately recovered solely from the observations $\{\mathbf{X}, \mathbf{t}, y\}$.*

**Assumption 6** (Proxy representation) *With proxies $\mathbf{X}$, there exists some representations $E(\mathbf{X}|\mathbf{t})$ (briefly, $E(\mathbf{X})$) such that $E(\mathbf{X}) \sim P(\mathbf{U})$, which means $Y(\mathbf{t}) \perp \mathbf{U} \mid E(\mathbf{X})$ for a potential outcome with the specific treatments $\mathbf{t}$.*

## 4 ESTIMATION

### 4.1 CONTRASTIVE REGULARIZER

Based on Assumption 6, the latent representation $\mathbf{U}$ will be obtained when approximating the representation $E(\mathbf{X})$. Existing methods (Louizos et al., 2017; Bica et al., 2020) address this problem by VAE (Kingma et al., 2019), GAN (Goodfellow et al., 2020) etc. They all rely on strong prior of the density form of $\mathbf{U}$. In this paper, inspired by Eq.(5), we propose a novel contrastive learning model to preserve $\mathbf{U}$ and eliminate $\Delta_{\epsilon_1}$ from data without explicit distribution prior.

**Contrastive Learning** There exists two functions $f \in \mathcal{F}$ and $g \in \mathcal{G}$, which encode $\mathbf{X}$ representations $f(\mathbf{X}, \epsilon_1)$ and $g(\mathbf{T}, \epsilon_2)$, satisfying $s(f(\mathbf{X}_i, \epsilon_1), g(\mathbf{T}_i, \epsilon_2)) >> s(f(\mathbf{X}_j, \epsilon_1), g(\mathbf{T}_i, \epsilon_2))$, where $i \neq j$. $s(\cdot, \cdot)$ is a function that measures the similarity between representations.

Contrastive learning approximates the latent representations by constructing contrastive samples (similar and dissimilar instances), by which similar instances are closer in the projection space, while dissimilar instances are further away in the projection space to maximize the lower bound of the mutual information.

Under CTP setting, even if we can control the observed proxies $\mathbf{X}$, the spurious association derived from $\mathbf{U}$ still can not be completely eliminated based on traditional representation algorithms. Therefore, we no longer rely on the representation balancing algorithm to cut off the relationship between $\mathbf{X}$ and $\mathbf{T}$ (even if we do, we cannot guarantee accurate estimation). Instead, we propose to strengthen the association between $\mathbf{X}$ and $\mathbf{T}$ using contrastive learning (Jaiswal et al., 2020) to model proxy representation $E(\mathbf{X})$ with neural network $E(\cdot)$ to represent the information from $\mathbf{U}$.

The essential part for the contrastive approach is the contrastive pairs for modeling representations. Inspired by Arbour et al. (2021); Li et al. (2020), we construct contrastive pairs with no need of discretizing the treatments in causal inference: $\mathbf{X}$ and $\mathbf{T}$ in original sample as positive pairs $\{(\mathbf{X}_i, \mathbf{T}_i)\}$ and $\mathbf{X}$ and $\mathbf{T}$ in permuted sample (shuffle $\mathbf{X}$ and $\mathbf{T}$ of data to obtain the permuted data) as negative pairs $\{\mathbf{X}_j, \mathbf{T}_i\}$. Then, as Fig. 2 shown, we set $E(\mathbf{X}|\mathbf{t}) = s(f(\mathbf{X}), g(\mathbf{t}))$ and adopt the NLL (Negative Log-Likelihood) loss (Chen et al., 2020) to design a novel contrastive loss to model $\mathbb{P}(\mathbf{U}|\mathbf{t}, \mathbf{X})$:

$$\ell_{Cr}(f(\mathbf{X}), g(\mathbf{t})) = -\log \frac{e^{(s(f(\mathbf{X}_i), g(\mathbf{t})))}}{\sum_{j=1}^{N} e^{(s(f(\mathbf{X}_j), g(\mathbf{t})))}}, \tag{6}$$

where $s(f(\mathbf{X}), g(\mathbf{t}))$ denotes the cosine similarity $\frac{f(\mathbf{X}) \cdot g(\mathbf{t})}{\|f(\mathbf{X})\|\|g(\mathbf{t})\|}$. In contrastive aspect, representations $\{g(\mathbf{T})\}$ are queries and representations $\{f(\mathbf{X})\}$ are keys. For a query $g(\mathbf{T}_i)$, the positive key is $f(\mathbf{X}_i)$ of the sample $i$ and the cosine similarity $s(f(\mathbf{X}_i), g(\mathbf{T}_i))$ value in the numerator in Eq. (6) is high. In contrast, the representation $f(\mathbf{X}_j)$ is the negative key of $g(\mathbf{T}_i)$ where $i \neq j$ and the

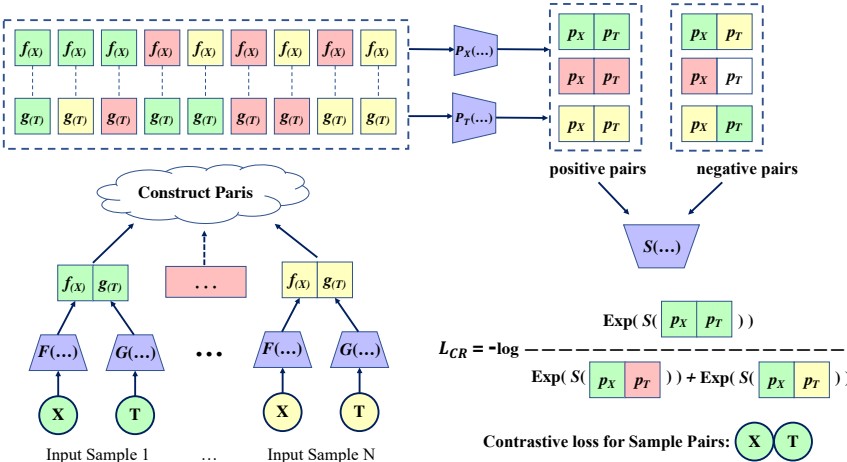

Figure 2: Contrastive regularizer. The covariates $\mathbf{X}$ are transformed to $f(\mathbf{X})$ via MLPs $\mathcal{F}$. In practice (Chen et al., 2020), the representation $f(\mathbf{X})$ is not directly constrained by contrastive loss. $f(\mathbf{X})$ transforms to $p_{\mathbf{X}}$ through projection head $P_X$. The treatments $\mathbf{T}$ are operated in a similar way. $p_{\mathbf{T}}$ and $p_{\mathbf{X}}$ are constrained by $\ell_{Cr}(f(\mathbf{X}), g(\mathbf{T}))$. For the sake of brevity, we use $f(\mathbf{X})$ and $g(\mathbf{T})$ in the context to represent $p_{\mathbf{X}}$ and $p_{\mathbf{T}}$.

cosine similarity $s(f(\mathbf{X}_j), g(\mathbf{T}_i))$ value in the denominator of Eq. (6) is low. Constrained by $\ell_{Cr}$, we capture the proxy representations $\{f(\mathbf{X}), g(\mathbf{T})\}$ from the data $\{\mathbf{X}, \mathbf{T}\}$.

In this section, we propose to strengthen association between treatments $\mathbf{T}$ and covariates $\mathbf{X}$ to recover unmeasured common causes $\mathbf{U}$ using $\ell_{Cr}$, which has two responsibilities: (1) strengthen the association between $\mathbf{X}$ and $\mathbf{T}$ in the same sample, (2) constrain the representation space using $\mathbf{X}$ and $\mathbf{T}$ in the permuted samples. Benefiting from contrastive learning, these two responsibilities complete each other. With contrastive learning constraints, learned representation $E(\mathbf{X})$ refuse the information of $\epsilon_1$ and maintain the information of $\mathbf{U}$, which means we eliminate the error item $\Delta_{\epsilon_1}$ in Eq.(5). Next, we consider the error term $\Delta_T$ in Eq.(5).

## 4.2 RANK WEIGHTING

### 4.2.1 MULTIPLE TREATMENT SCORE WEIGHT

Estimating ADRF given proxy representation, existing methods usually apply the balancing methods [1] to approximate the density $\mathbb{P}(\mathbf{T}|\mathbf{U})$ for balancing score weights. That is, adopting the inverse of the approximated $\hat{\mathbb{P}}(\mathbf{T}|\mathbf{U})$ as the sample weights: $W_i = \frac{1}{\hat{\mathbb{P}}(\mathbf{T}_i|\mathbf{U}_i)}$. However, $\hat{\mathbb{P}}(\mathbf{T}|\mathbf{U})$ is sensitive to correct specified and nearly can not be estimated accurately, especially for high-dimensional $\mathbf{U}$ and multi-dimensional continuous $\mathbf{T}$.

To approximate the $\mathbb{P}(\mathbf{T}|\mathbf{U})$ under CTP setting, we adopt a mixture density network (MDN, which uses neural network to learn the Gaussian mixture model, (Bishop, 1994)) to model $\hat{\mathbb{P}}(\mathbf{T}|\mathbf{U}) = \prod_{q=1}^{Q} \hat{\mathbb{P}}(t^q \mid \mathbf{U})$. As the Fig. 3 shown, we apply MDN to approximate $\hat{\mathbb{P}}(\mathbf{T}|\mathbf{U})$ and obtain the sample weight $W_i$ as

$$W_i = \frac{1}{\hat{\mathbb{P}}(\mathbf{T}_i|\mathbf{U}_i)}, \hat{\mathbb{P}}(\mathbf{T}_i|\mathbf{U}) = \sum_{k=1}^{K} \alpha_{\mathrm{k}} \mathcal{N}\left(\mathbf{T}_i \mid \mu_{\mathrm{k}}, \mathbf{\Sigma}_{\mathrm{k}}^2\right) \tag{7}$$

---

[1]Given that both matching and stratification methods can be considered as special cases of weighted methods, and that the first two methods require discretization of $\mathbf{T}$ when it is continuous values, this paper focuses on weighting methods.

The loss function of $R_w$ is:

$$l_{Rw} = -\frac{1}{n} \sum_{i=1}^{n} \log \left\{ \sum_{k=1}^{K} \alpha_k \mathcal{N} \left( \mathbf{T}_i \mid \mu_k, \mathbf{\Sigma}_k^2 \right) \right\}. \tag{8}$$

where $K$ is the number of sub-Gaussian models $\mathcal{N}(\cdot)$ in the Gaussian mixture model, $(\mu_k, \mathbf{\Sigma}_k)$ is the mean vector and covariance matrix of the $k$th sub-Gaussian model, and $\alpha_k$ is the probability that the observation belongs to the $k$th sub-Gaussian model.

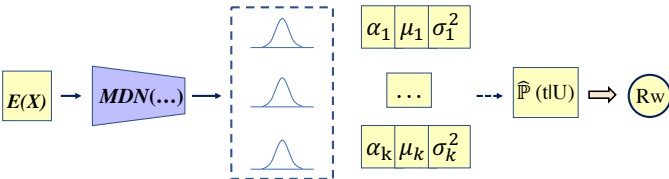

Figure 3: Rank weighting. The proxy representation $E(\mathbf{X})$ are transformed to Gaussian mixture distribution with $K$ Gaussian submodels $\mathcal{N}(\alpha_k, \mu_k, \mathbf{\Sigma}_k^2)$ via MLPs MDN. Then we infer the estimated density $\hat{\mathbb{P}}(\mathbf{t}|\mathbf{U})$ and sort it in descending order to get the rank weight Rw.

### 4.2.2 RANK WEIGHT

However, the sample weight $W$ from $\hat{\mathbb{P}}(\mathbf{T}|\mathbf{U})$ still suffers from extreme values problem. Although recent methods (Fong et al., 2018; Vegetabile et al., 2021; Huling et al., 2021) propose to clip the score value or optimize balance weights directly, they still face a dilemma when the data gets more complex, especially, under multi-continuous treatment setting. So a balancing method that have no extreme values and adapted to unobserved confounders is urgently needed. Therefore, based on the proxy representation learned above, the core contribution of this paper is the rank weights for more effective balancing weighting.

Motivated by the balancing score problem, we normalize the IPW weights obtained from Eq. (7) $\{W_i\}_{i=1}^n \in [0,1]$ and sort them in descending order. We record their $rank$ (the order in sorted data) as $R_i \in \mathbb{N}$ and the difference between adjacent $W$ as $stride$ $\delta \in \mathbb{R}$. We define the IPW as $\xi(R, \delta) = \frac{1}{\hat{\mathbb{P}}(\mathbf{t}|\mathbf{U})}$. It's clear that when sample size $n$ is limited, the large $\delta$ causes extreme values and when $n \to \infty$, $\delta \to 0$, so we define the form of rank weight as $\tilde{\xi}(R)$ and propose

**Proposition 1** *There exists some rank weight $\tilde{\xi}(R)$ that when $n \to \infty$, $\frac{e^{-\tilde{\xi}(R)}}{Z} \to \frac{1}{\mathbb{P}(\mathbf{t}|\mathbf{U})}$, where $Z = \int e^{-\tilde{\xi}(R)}$ is the normalizing constant of $\tilde{\xi}(R)$.*

The Proposition 1 shows that the causal effect estimation with rank weight approximate to the unbiased estimation of causal effect. We detail the definition and proposition in Appendix. When $n$ is limited, to eliminate the stride, we set it to a constant $\delta = \frac{1}{n}$, and obtain the rank weight

$$\xi(\hat{R}_i) \approx R_i. \tag{9}$$

The Eq. (9) shows that Rw method is adapted to Cr and can be applied to data of any dimension because it only depends on the rank information of weights. when $n \to \infty$, the rank weight approximates to $\frac{1}{\mathbb{P}(\mathbf{t}|\mathbf{U})}$. With limited data samples, the rank weight don't rely on specified models and address extreme values problem.

Fig. 3 shows the process of rank weighting: After training of MDN [2], we inference $\hat{\mathbb{P}}(\mathbf{T}|\mathbf{U}) = \sum_{k=1}^{K} \alpha_k \mathcal{N} \left( \mathbf{T} \mid \mu_k, \sigma_k^2 \right)$. Then we sort $\hat{\mathbb{P}}(\mathbf{T}|\mathbf{U})$ in descending order and get the rank weight $Rw_i = \tilde{\xi}(R_i)$. Then the $\Delta_T$ of Eq.(5) can be eliminated by weighted regression with rank weight.

---

[2] Note that the rank weight is not only adapted to MDN, it can be applied to any balancing weights.

### 4.3 CRNET

Combining the Contrastive regularizer and Rank weight, we propose a neural network framework CRNet to estimate ADRF under CTP setting to eliminate $\Delta_Y$. As Fig. 4 shown, the overall CRNet

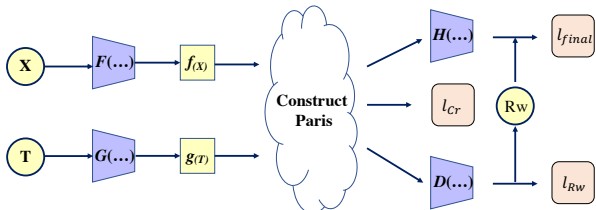

Figure 4: CRNet. For training procedure, the proxy representation $E(\mathbf{X}|\mathbf{t}) = \{f(\mathbf{X}), g(\mathbf{t})\}$ constrained by contrastive loss $l_{Cr}(\mathbf{X}, \mathbf{t})$ are concatenated and input to MLPs $H$ and MDN $D$ to obtain the estimated outcome $\hat{Y}$ and the rank weights $Rw$. The final objective is to minimize the weighted loss in Eq. 10. For inference procedure, the estimated ADRF is obtained by $h(f(\mathbf{X}), g(\mathbf{t}))$.

architecture contains three components: (1) a contrastive regularizer, which contains two MLPs heads that encode proxy and the observed treatments into representations $\{f(\mathbf{X}), g(\mathbf{T})\}$. (2) A sample weight learner named tank weighting. This module optimizes the rank weights on $\ell_{Rw}$ in Eq.(8) using the representations $\{f(\mathbf{X}), g(\mathbf{T})\}$. (3) A base MLPs encoder $H$ that concatenates $f(\mathbf{X})$ and $g(\mathbf{T})$ and transforms them as the estimated $\hat{Y}$ to approximate the observed $Y$ by the weighted regression loss $\ell_{final}(W, \mathbf{X}, \mathbf{T}, Y)$.

Combining $\ell_{Cr}(\mathbf{X}, \mathbf{T})$ and $\ell_{Rw}(W)$, the final loss is defined as:

$$\ell_{final} = \sum_{i=1}^{N} Rw_i * (Y_i - \hat{Y}_i)^2 + \alpha * \ell_{Er}(\mathbf{X}, \mathbf{T}) + \beta * \ell_{Rw}(W), \tag{10}$$

where $Rw_i$ is the rank weight optimized by $\ell_{Rw}(W)$. $\alpha$ and $\beta$ are the hyperparameters of Cr and Rw, respectively.

## 5 EXPERIMENTS

To evaluate the performance of CRNet for CTP problem, we compare 6 statistical methods and 5 deep-based methods as baselines in ten simulation data and four semi-synthesis data from *IHDP & News*. All experiments are implemented using PyTorch (Paszke et al., 2019) on Intel(R) Xeon(R) Gold 6240 CPU @ 2.60GHz.

**Baselines** We compare our model with following baselines: For statistical methods, we use (1) **CausalForest** (Wager & Athey, 2018), a random forest algorithm for causal inference. (2) **Bart** (Hill, 2011; Chipman et al., 2010), Bayesian Additive Regression Trees for causal inference. (3) **GPS** (Hirano & Imbens, 2004), a generalized propensity score for continuous treatments. (4)**CBGPS** (Fong et al., 2018), a generalized covariate balancing propensity score (Imai & Ratkovic, 2014) for continuous treatments. (5)**EB**, a continuous treatment version of entropy balancing method (Hainmueller, 2012). (6)**DCOWS** (Huling et al., 2021), a balancing method based on the distance covariance (Székely & Rizzo, 2009). For representation based methods, we apply (7) **NN**, a neural network with fully MLPs. (8) **MDN** (Bishop, 1994), a mixture density network for modelling the density. (9) **DRNet** (Schwab et al., 2020), a multi-head deep model stratified according to $\mathbf{T}$, we use a modified version (Nie et al., 2021) for estimating ADRF. (10) **VCNet** (Nie et al., 2021), a deep model which considers $\mathbf{T}$ as a varying coefficient. (11) **CEVAE** (Louizos et al., 2017), a VAE-based model to constrain the representation of covariates.

**Datasets** We evaluate the performance of CRNet in ten simulation data and four semi-synthesis data. For simulation experiments, we design 10 simulation datasets and named five of them $Data_X\_D_\mathbf{T}\_D_\mathbf{X}$ (e.g., $Data_X\_1\_5$ means a simulation with 1 treatment, 5 covariates and no unobserved confounders). We name the other 5 of them $Data_U\_D_\mathbf{T}\_D_\mathbf{X}$ with unobserved confounders.

We also conduct 4 semi-synthetic experiments on 2 real-world datasets: IHDP[3] and News[4]. IHDP contains 747 observations on 25 covariates. Following Schwab et al. (2020), we sample 5000 samples with 2870 covariates from News. The semi-synthetic experiment on IHDP is named as IHDP_$D_{\mathbf{T}}$. The other 3 semi-synthetic experiments on News are named as News_$D_{\mathbf{T}}$. For detailed descriptions of datasets, models, and hyperparameters, see Appendix.

**Metrics**  For all experiments, we perform 30 replications ($E = 30$) to report the average mean squared error (MSE) and the standard deviations (SD) of the average dose-response function estimation. For correlation measurement, we adopt distance correlation (dCor) to evaluate the quality of the proxy representation $E(\mathbf{X})$. $\mathrm{dCor}(X, Y) = \frac{\mathrm{dCov}(X,Y)}{\sqrt{\mathrm{d\,Var}(X)\,\mathrm{d\,Var}(Y)}}$ with $\mathrm{Var}(X) = \mathrm{dCov}(X, X)$, $\mathrm{dCov}(\mathbf{X}, \mathbf{T}) := \frac{1}{n^2} \sum_{i=1}^{n} \sum_{j=1}^{n} \mathbf{A}_{i,j} \mathbf{B}_{i,j}$, where $\mathbf{A}_{i,j} := \mathbf{a}_{i,j} - \bar{\mathbf{a}}_{i\cdot} - \bar{\mathbf{a}}_{\cdot j} + \bar{\mathbf{a}}_{\cdot\cdot}$, $\mathbf{a}_{i,j} = \|\mathbf{X}_i - \mathbf{X}_j\|_2$, $\bar{\mathbf{a}}_{i\cdot} = \frac{1}{n} \sum_{j=1}^{n} \mathbf{a}_{i,j}$, $\bar{\mathbf{a}}_{\cdot j} = \frac{1}{n} \sum_{j=1}^{n} \mathbf{a}_{i,j}$, $\bar{\mathbf{a}}_{\cdot\cdot} = \frac{1}{n^2} \sum_{i,j=1}^{n} \mathbf{a}_{i,j}$. And the form of $B_{i,j}$ is similar.

**Experimental results on simulation datasets**  To assess the performance of CRNet, we conduct simulation experiments increasing the dimensions of treatments and proxies. As Table 1 shown, all methods except GPS perform well in the low-dimensional $Data_U\_1\_5$. All baselines fails when treatments are multiple in $Data_U\_2\_200$ and $Data_U\_5\_200$. Increasing the dimension of treatments as $Data_U\_2\_200$ and $Data_U\_5\_200$, we found that CEVAE, which performs well in low dimensions, fails in convergence, which has also been demonstrated in Rissanen & Marttinen (2021). The performance of CRNet outperforms others across different settings. We further verify the effectiveness of Cr module in CRNet below.

Table 1: Results (MSE±SD) on simulation $Data_U\_D_{\mathbf{T}}\_D_{\mathbf{X}}$

| E=30 | $Data_U\_1\_5$ | $Data_U\_1\_50$ | $Data_U\_1\_200$ | $Data_U\_2\_200$ | $Data_U\_5\_200$ |
|---|---|---|---|---|---|
| Causal Forest | $3.962 \pm 0.9519$ | $12.22 \pm 15.530$ | $76.56 \pm 79.527$ | $82.30 \pm 82.395$ | $96.89 \pm 89.262$ |
| Bart | $3.905 \pm 0.9021$ | $9.445 \pm 11.109$ | $129.9 \pm 83.055$ | $127.8 \pm 54.624$ | $154.3 \pm 66.021$ |
| GPS | $42.25 \pm 3.0993$ | $65.67 \pm 53.597$ | $1030. \pm 119.94$ | $1110. \pm 140.00$ | $1112. \pm 140.51$ |
| NN | $1.914 \pm 0.5765$ | $2.542 \pm 2.8093$ | $22.78 \pm 11.503$ | $26.71 \pm 9.8489$ | $22.87 \pm 8.8573$ |
| DRNet | $1.982 \pm 0.6883$ | $3.422 \pm 1.6802$ | $17.24 \pm 9.7990$ | $21.77 \pm 6.9279$ | $23.95 \pm 9.2497$ |
| VCNet | $1.457 \pm 1.4142$ | $3.502 \pm 3.7841$ | $12.95 \pm 17.753$ | $27.22 \pm 17.365$ | $25.28 \pm 15.617$ |
| CEVAE | $0.944 \pm 0.1382$ | $3.308 \pm 0.2950$ | $7.309 \pm 0.6142$ | $21.44 \pm 1.2480$ | $25.04 \pm 1.9198$ |
| CRNet | $\mathbf{0.865 \pm 0.2212}$ | $\mathbf{1.360 \pm 1.5138}$ | $\mathbf{6.651 \pm 9.4864}$ | $\mathbf{11.79 \pm 10.047}$ | $\mathbf{14.79 \pm 15.678}$ |

**Experimental results on the effectiveness of Cr block**  In the setting with multiple continuous treatments and proxies, we propose the Cr to model E($\mathbf{X}$) to hold onto the information from unobserved confounders. Practitioners use representation-based approaches to map proxies into a low-dimension representation space which will lose information predictive of the predicted treatment variable. As shown in Fig. 5(a) [5], the correlation between $E(\mathbf{X})$ and $\mathbf{T}$ from conventional methods is still weak. To retain the information predictive, Cr regularize the proxy representation $E(\mathbf{X})$ by contrastive learning, the correlation from CRNet and CRNet(ft) [6] is strong. In the experiment of Fig. 5(b), we first train a $\mathbf{U}$-to-$\mathbf{T}$ prediction network to obtain the representation $\mathbf{T}(\mathbf{U})$ to represent the relationship between $\mathbf{T}$ and $\mathbf{U}$. X_T(U) denotes the dCor of $\mathbf{X}$ and $\mathbf{T}(\mathbf{U})$. DRNet refers to the dCor between $\mathbf{T}(\mathbf{U})$ and representation $f(\mathbf{X})$. Others are operated similarly. It demonstrates that Cr successfully regularized the representation $E(\mathbf{X})$ between $\mathbf{X}$ and $\mathbf{T}$ and other methods not.

**Experimental results on the effectiveness of Rw block**  Our downstream block for estimation is Rw. It is reliable iff the proxy representation is accurately measured. To evaluate the performance of Rw block, we conduct 5 simulation experiments with no unobserved confounders. As shown in Table 2, all experiments use the same backbone $\mathbf{NN}$. In all experiments, GPS and MDN which direct model the density of $\mathbb{P}(\mathbf{T}|\mathbf{X})$ induce excessive bias in ADRF estimation. The direct rank weight without Cr performs well in all experiments. And with Cr, our rank weighting method outperforms other weighting methods. It demonstrates that our CRNet is state-of-the-art even with no unobserved confounders.

---

[3]https://www.fredjo.com

[4]https://paperdatasets.s3.amazonaws.com/news.db

[5]$\mathbf{X}\_\mathbf{T}$ refers the dCor between the proxies $\mathbf{X}$ and treatments $\mathbf{T}$.

[6]CRNet(ft) means the correlation between $f(\mathbf{X})$ and $g(\mathbf{T})$.

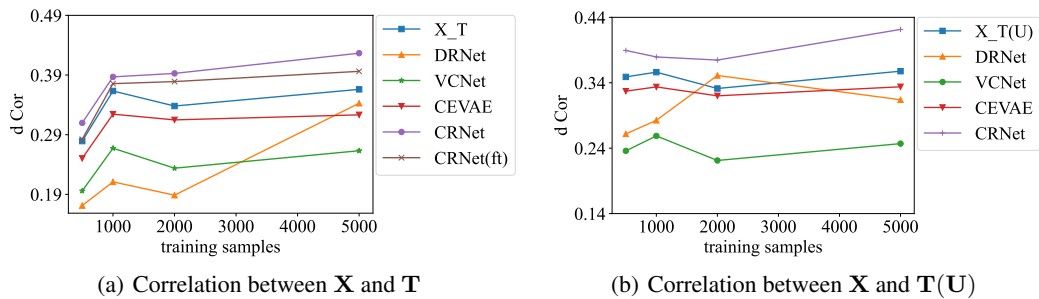

(a) Correlation between $\mathbf{X}$ and $\mathbf{T}$      (b) Correlation between $\mathbf{X}$ and $\mathbf{T(U)}$

Figure 5: Correlation of treatments, unobserved confounders, and covariates. In both figures, the abscissa represents the sample size, and the ordinate represents the value of dCor.

Table 2: Results (MSE±SD) on simulation $Data_X\_D_\mathbf{T}\_D_\mathbf{X}$

| E=30 | $Data_X\_1\_5$ | $Data_X\_1\_50$ | $Data_X\_1\_200$ | $Data_X\_2\_200$ | $Data_X\_5\_200$ |
|---|---|---|---|---|---|
| NN | $0.317 \pm 0.2181$ | $2.012 \pm 2.3314$ | $5.440 \pm 5.6123$ | $5.606 \pm 7.3323$ | $12.75 \pm 20.344$ |
| +GPS | $1.055 \pm 1.2795$ | $13.31 \pm 25.384$ | $36.58 \pm 62.489$ | $28.20 \pm 33.284$ | $47.64 \pm 70.285$ |
| +CBGPS | $0.323 \pm 0.2911$ | $1.353 \pm 1.8173$ | $6.196 \pm 6.9755$ | $5.472 \pm 6.4689$ | $8.019 \pm 5.2724$ |
| +EB | $0.552 \pm 0.9203$ | $2.933 \pm 4.0231$ | $6.398 \pm 5.6984$ | $4.217 \pm 5.9709$ | $8.210 \pm 3.0670$ |
| +DCOWS | $0.308 \pm 0.2841$ | $1.680 \pm 1.6652$ | $6.048 \pm 5.3401$ | $4.727 \pm 6.2233$ | $9.490 \pm 7.9573$ |
| +MDN | $0.589 \pm 1.6048$ | $13.11 \pm 45.473$ | $99.26 \pm 246.86$ | $12.03 \pm 18.670$ | $18.44 \pm 19.405$ |
| +Rw | $0.268 \pm 0.2762$ | $1.171 \pm 0.8979$ | $3.436 \pm 4.0289$ | $3.935 \pm 4.1624$ | $7.500 \pm 3.9411$ |
| +Cr+Rw | $\mathbf{0.105 \pm 0.0308}$ | $\mathbf{1.170 \pm 1.0302}$ | $\mathbf{3.043 \pm 4.9246}$ | $\mathbf{1.915 \pm 5.1747}$ | $\mathbf{5.654 \pm 2.8299}$ |

**Experimental results on real-world datasets** We further verify the performance of CRNet in real-world datasets IHDP & News. As shown in Table 3, the traditional methods Bart and Causal Forest cannot estimate the treatment effect accurately and suffer from the high-dimensional proxy imbalanced between different treatments. CRNet obtain a high-quality representation $E(\mathbf{X})$ and retain predictive information of the predicted treatments in representation using contrastive regularizer, but other deep-based methods fails to capture the rich information between high-dimensional covariates and treatments. Therefore, CRNet shows robust performance and achieves the state-of-the-art in all real-world experiments.

Table 3: Results (MSE±SD) on semi-simulation Real-Data$\_D_\mathbf{T}$

| E=30 | IHDP_1 | News_1 | News_2 | News_5 |
|---|---|---|---|---|
| Causal Forest | $0.576 \pm 0.5651$ | $24.60 \pm 4.7884$ | $20.08 \pm 4.0395$ | $21.23 \pm 4.7864$ |
| Bart | $0.514 \pm 0.3728$ | $6.200 \pm 12.018$ | $11.83 \pm 3.2380$ | $22.25 \pm 17.902$ |
| GPS | $3.198 \pm 15.148$ | $1.603 \pm 0.2932$ | $2.293 \pm 0.7984$ | $6.085 \pm 1.1495$ |
| NN | $0.774 \pm 1.5413$ | $0.958 \pm 0.1148$ | $2.435 \pm 0.2597$ | $7.435 \pm 1.0865$ |
| DRNet | $1.485 \pm 1.4620$ | $3.442 \pm 1.3222$ | $6.486 \pm 4.2931$ | $10.18 \pm 2.8848$ |
| VCNet | $0.626 \pm 0.7126$ | $8.563 \pm 9.4974$ | $7.578 \pm 9.2410$ | $10.56 \pm 6.4256$ |
| CEVAE | $1.935 \pm 0.7765$ | $1.142 \pm 0.0762$ | $3.289 \pm 0.1860$ | $9.660 \pm 0.4348$ |
| CRNet | $\mathbf{0.351 \pm 0.2122}$ | $\mathbf{0.867 \pm 0.2193}$ | $\mathbf{2.115 \pm 0.6143}$ | $\mathbf{5.081 \pm 0.7098}$ |

## 6 CONCLUSION

For CTP problem, we formulate the estimation error into three terms from recovery of unobserved confounder, treatment assignment and approximation of outcome. We propose the contrastive regularizer to constrain the proxy representation in representation space for the bias from recovery of unobserved confounder. Based on Cr, we propose a rank weighting method to eliminate the extreme values problem and alleviate the sensitivity problem to model misspecified in treatment assignment model. Combining Cr and Rw, we elaborate a CRNet adapted to CTP problem to reduce the outcome approximation bias. CRNet achieves the state-of-the-art performance in estimating ADRF of both synthetic and semi-synthetic data.

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

# A APPENDIX

## A.1 IDENTIFICATION

Given latent unconfoundedness and proxy assumption, we know that the causal effect is not identified conditioning on $\mathbf{X}$. Because $\epsilon_1 \not\perp \mathbf{U}|\mathbf{X}$, then $Y(\mathbf{t}) \not\perp \mathbf{T}|\mathbf{X}$. We also show this problem in ADRF adjustment formula. The true ADRF is identified as

$$ADRF^* = \mathbb{E}[Y(\mathbf{t})]] = \mathbb{E}_{\mathbf{U}}[\mathbb{E}[Y(\mathbf{t}) \mid \mathbf{U}]]] = \mathbb{E}_{\mathbf{U}}[\mathbb{E}[Y(\mathbf{t}) \mid \mathbf{T} = \mathbf{t}, \mathbf{U}]] = \mathbb{E}_{\mathbf{U}}[\mathbb{E}[Y \mid \mathbf{T} = \mathbf{t}, \mathbf{U}]] \tag{11}$$

When proxy exists, ADRF is identified as

$$\hat{ADRF} = \mathbb{E}[Y(\mathbf{t})]] = \mathbb{E}_{\mathbf{X}}[\mathbb{E}[Y(\mathbf{t}) \mid \mathbf{X}]]] = \mathbb{E}_{\mathbf{X}}[\mathbb{E}[Y(\mathbf{t}) \mid \mathbf{T} = \mathbf{t}, \mathbf{X}]] = \mathbb{E}_{\mathbf{X}}[\mathbb{E}[Y \mid \mathbf{T} = \mathbf{t}, \mathbf{X}]]$$
$$\neq \mathbb{E}_{\mathbf{U}}[\mathbb{E}[Y \mid \mathbf{T} = \mathbf{t}, \mathbf{U}]] = ADRF^*. \tag{12}$$

It is clear that the adjustment formula of ADRF from proxy is different from that of true ADRF because $Y(\mathbf{t}) \not\perp \mathbf{T}|\mathbf{X}$, it will induce the recovery error $\Delta_{\epsilon_1}$ in the ADRF estimation phase if the unmeasured confounder $\mathbf{U}$ is not correct specified. The performance of using proxy to estimate ADRF depends on the degree of recovery of $\mathbf{U}$.

## A.2 PROOF OF EQUATION (4)

We set $\hat{ADRF}$ as baseline and define the estimation error as

$$\begin{aligned}
\Delta = ADRF^* - \hat{ADRF} &= \mathbb{E}[Y_i(\mathbf{T}_i = \mathbf{t})] - \mathbb{E}[\phi(\mathbf{X}_i|\mathbf{t})] \\
&= \mathbb{E}[\phi^*(\mathbf{U}, do(\mathbf{t}))] - \mathbb{E}[\hat{\phi}(\mathbf{U}_i|\mathbf{t}) + \hat{h}(\epsilon_{1i}|\mathbf{t})] \\
&= \mathbb{E}[\phi^*(\mathbf{U}, do(\mathbf{t}))] - \mathbb{E}[\hat{\phi}(\mathbf{U}_i|\mathbf{t})] - \mathbb{E}[\hat{h}(\epsilon_{1i}|\mathbf{t})] \\
&= \mathbb{E}[\phi^*(\mathbf{U}, do(\mathbf{t}))] - \mathbb{E}[\phi^*(\mathbf{U}|\mathbf{t})] + \mathbb{E}[\phi^*(\mathbf{U}|\mathbf{t})] - \mathbb{E}[\hat{\phi}(\mathbf{U}_i|\mathbf{t})] - \mathbb{E}[\hat{h}(\epsilon_{1i}|\mathbf{t})] \\
&= \mathbb{E}[\phi^*(\mathbf{U}, do(\mathbf{t})) - \phi^*(\mathbf{U}|\mathbf{t})] + \mathbb{E}[\phi^*(\mathbf{U}|\mathbf{t}) - \hat{\phi}(\mathbf{U}_i|\mathbf{t})] - \mathbb{E}[\hat{h}(\epsilon_{1i}|\mathbf{t})]
\end{aligned} \tag{13}$$

## A.3 ANALYSIS FOR RANK WEIGHT

Based on the definition rank, we define the corresponding index function of $rank\ I(R_i) = i$. And we record the *stride*

$$\delta_{R_i} = \begin{cases} W_{I(R_i-1)} - W_i & 0 < R_i < n \\ W_i & R_i = 0 \end{cases} \tag{14}$$

as the difference between two adjacent weights of the sorted data. Then we can build a sequence model of $W$ as:

$$W_i = \begin{cases} W_{I(R_i-1)} - \delta_{R_i} & 0 < R_i < n \\ \delta_{R_i} & R_i = 0 \end{cases} \tag{15}$$

Supposing an extreme value example that the maximum weight is much larger than others, this is because there are many unmeasured weights between the largest weight and the second largest weight we obtained. Combining Eq.(15) and the continuity of the probability density, it is clear that the excessive $stride$ causes the extreme value of the weights. Moreover, the sample weights with large $stride$ will also cause the sensitivity to misspecified because slight misspecifed of the large $stride$ induce significant bias. Then is just using the $rank$ information enough to make a covariate balance? To answer this question, we formulate the model in Eq.(15) as:

$$W = \psi(R, I(R), \delta) = \xi(R, \delta), \tag{16}$$

where $W$ is the sample weight, $R$ is the *rank*, $I$ is the $index$ function of $R$ and $\delta$ is $stride$. We reduce the second line to third line because $I$ is a deterministic function of $R$.

Based on Eq.(14) we notice that in $\{W_i\}_{i=1}^{n} \in [0,1]$, if $n \to \infty$, then $stride \to 0$ because $W_{R_i} - W_{R_i-1} \to 0$. It is similar to the unnormalized density. Therefore, we formulate an IPW via Gibbs sampling: $\frac{1}{P(\mathbf{T}|\mathbf{U})} = \frac{e^{-W}}{Z} = \frac{e^{-\tilde{\xi}(R,\delta)}}{Z}$, where $Z = \int e^{-\xi(R,\delta)}$. We set $n \to \infty$, then $\frac{e^{-\tilde{\xi}(R))}}{Z} \to \frac{e^{-\xi(R,\delta)}}{Z} = \frac{1}{\mathbb{P}(\mathbf{t}|\mathbf{U})}$. So we propose that

**Proposition** *There exists some rank weight $\tilde{\xi}(R)$ that when $n \to \infty$, $\frac{e^{-\tilde{\xi}(R)}}{Z} \to \frac{1}{\mathbb{P}(\mathbf{t}|\mathbf{U})}$, where $Z = \int e^{-\tilde{\xi}(R)}$ is the normalizing constant of $\tilde{\xi}(R)$.*

The proposition and Eq.(16) show that when $n \to \infty$, using rank weight $\tilde{\xi}(R)$ is enough to make a covariate balance because the weights $Rw$ approximate to the IPW of density $\mathbb{P}(\mathbf{T}|\mathbf{U})$. It means when $n \to \infty$, the causal effect estimation with $Rw$ approximates to the unbiased estimation of causal effect (for unbiased estimation with IPW of $\hat{\mathbf{P}}(\mathbf{T}|\mathbf{U})$, see Imbens (2000)). And when the data size is limited, rank function $\tilde{\xi}(R)$ can effectively avoid the extreme value problem and alleviate the sensitivity to the model misspecified. So we direct eliminate $delta$ by setting $\delta = \frac{1}{n}$ to obtain $\hat{\tilde{\xi}}(R) \approx R_i$. The operation can be considered as enforce the distribution of $delta$ is Uniform, which is biased. But as $n \to \infty$, the obtained $\hat{\tilde{\xi}}(R) \approx R_i$ is approximating to the true IPW.

## A.4 EXPERIMENTAL DETAILS

The rules for defining symbols in this section are the same as in the main body. Please note that when the superscript is specified as $A^{p=2}$, it means the dimension of $\mathbf{A}$ is 2, and when it is not specified (e.g., $\mathbf{A}^2$), it means the power of $\mathbf{A}$ is 2.

### A.4.1 DETAILS ON DATASETS

The dataset split and dimension information corresponding to the data name are expressed in Table 4.

Table 4: Dataset description

|  | $N_{train}/N_{test}$ | $D_T$ | $D_X$ | $D_U$ | $D_Y$ |
|---|---|---|---|---|---|
| $Data_U\_1\_5$ | 1800/300 | 1 | 5 | 5 | 1 |
| $Data_U\_1\_50$ | 1800/300 | 1 | 50 | 50 | 1 |
| $Data_U\_1\_200$ | 1800/300 | 1 | 200 | 200 | 1 |
| $Data_U\_1\_200$ | 1800/300 | 2 | 200 | 200 | 1 |
| $Data_U\_1\_200$ | 1800/300 | 5 | 200 | 200 | 1 |
| $Data_X\_1\_5$ | 1800/300 | 1 | 5 | 0 | 1 |
| $Data_X\_1\_50$ | 1800/300 | 1 | 50 | 0 | 1 |
| $Data_X\_1\_200$ | 1800/300 | 1 | 200 | 0 | 1 |
| $Data_X\_1\_200$ | 1800/300 | 2 | 200 | 0 | 1 |
| $Data_X\_1\_200$ | 1800/300 | 5 | 200 | 0 | 1 |
| IHDP_1 | 672/75 | 1 | 25 | 25 | 1 |
| News_1 | 3150/1350 | 1 | 2870 | 20 | 1 |
| News_2 | 3150/1350 | 2 | 2870 | 20 | 1 |
| News_5 | 3150/1350 | 5 | 2870 | 20 | 1 |

**Synthetic datasets** we construct synthetic datasets following EB (Vegetabile et al., 2021). For all simulation datasets, the true covariates $\mathbf{U}^{p=1\cdots200}$ are constructed as: $\mathbf{U}^{p=1\cdots200} \sim \mathcal{N}(0,1)$.

For $Data_X\_1\_5$, $Data_X\_1\_50$, $Data_X\_1\_200$ datasets, $\mathbf{T}^{p=1}$ is constructed as:

$$\mathbf{T}^{p=1} = 0.5\mathcal{N}(3,1) + 0.5\mathcal{N}(6,0.5) + 1.5 * \sum_{p=1}^{p=3} \mathbf{X}^p$$

And $Y^{p=1}$ is constructed as:

$$Y^{p=1} = \frac{1}{e^{-\sum_{p=1}^{p=2} \mathbf{T}^p}} + e^{\mathbf{X}^{p=1}} + 2.1 * \mathbf{X}^{p=2} + 2.2 * \mathbf{X}^{p=3}$$

$$+ 2.3 * \mathbf{X}^{p=4} + \mathbf{X}^{p=5} + 4.0 * \sum_{p=151}^{p=200} \mathbf{X}^p + \mathbb{I}(D_{\mathbf{T}==5} * (3 * \cos(\sum_{p=3}^{p=5} \mathbf{T}^p))$$

For $Data_U\_1\_5$, $Data_U\_1\_50$, $Data_U\_1\_200$ datasets, $\mathbf{T}^{p=1}$ is constructed as:

$$\mathbf{T}^{p=1} = 0.5\mathcal{N}(3,1) + 0.5\mathcal{N}(6,0.5) + 1.5 * \sum_{p=1}^{p=3} \mathbf{U}^p + 0.5 * \sum_{p=151}^{p=D_{\mathbf{U}}} \mathbf{U}^p.$$

And $Y^{p=1}$ is constructed as:

$$Y^{p=1} = \frac{1}{e^{-\sum_{p=1}^{p=2} \mathbf{T}^p}} + e^{\mathbf{U}^{p=1}} + 2.1 * \mathbf{U}^{p=2} + 2.2 * \mathbf{U}^{p=3} + 2.3 * \mathbf{U}^{p=4}$$

$$+ \mathbf{U}^{p=5} + \mathbb{I}(D_{\mathbf{U}} > 5) * \frac{D_{\mathbf{U}}}{50} - \mathbb{I}(D_{\mathbf{U}} == 200) * 2$$

$$+ \frac{100}{D_{\mathbf{U}} + 5} * \sum_{p=6}^{p=151} \mathbf{U}^p + 4.0 * \sum_{p=151}^{p=200} \mathbf{U}^p - 1$$

where $D_{\mathbf{U}}$ is the dimension of $\mathbf{U}$ and $\mathbb{I}$ is the indicator function.

For $Data_U\_2\_200$ and $Data_U\_5\_200$ datasets, $\mathbf{T}^{p=1\cdots5}$ is constructed as:

$$\mathbf{T}^{p=1} = 0.5\mathcal{N}(3,1) + 0.5\mathcal{N}(6,0.5) + 1.5 * \sum_{p=1}^{p=4} \mathbf{U}^p + 0.4 * \sum_{p=151}^{p=D_{\mathbf{U}}} \mathbf{U}^p,$$

$$\mathbf{T}^{p=2} = \mathcal{N}(4,1) + 1.5 * \mathbf{U}^{p=5}.$$

$$\mathbf{T}^{p=3\cdots5} = \mathcal{N}(p,0.5) + \sum_{q=100}^{100+p} \mathbf{U}^q.$$

And $Y^{p=1}$ is constructed as:

$$Y^{p=1} = \frac{1}{e^{-\sum_{p=1}^{p=2} \mathbf{T}^p}} + e^{\mathbf{U}^{p=1}} + 2.1 * \mathbf{U}^{p=2} + 2.2 * \mathbf{U}^{p=3} + 2.3 * \mathbf{U}^{p=4}$$

$$+ \mathbf{U}^{p=5} + \mathbb{I}(D_{\mathbf{U}} > 5) * \frac{D_{\mathbf{U}}}{50} - \mathbb{I}(D_{\mathbf{U}} == 200) * 2$$

$$+ \frac{100}{D_{\mathbf{U}} + 5} * \sum_{p=6}^{p=151} \mathbf{U}^p + 4.0 * \sum_{p=151}^{p=200} \mathbf{U}^p$$

$$+ \mathbb{I}(D_{\mathbf{T}} == 5) * (0.1 * \sum_{p=3}^{p=5} \mathbf{T}^p + 2) - 1$$

The observed covariates $\mathbf{X}^{p=1\cdots200}$ of $Data_U\_1\_5$, $Data_U\_1\_50$, $Data_U\_1\_200$, $Data_U\_2\_200$ and $Data_U\_5\_200$ are formulated as

$$\mathbf{X}^{p=1\cdots5} = \mathbf{U}^p + linespace(0, \frac{p}{10}, N_{\mathbf{X}})$$

$$\mathbf{X}^{p=5\cdots200} = \mathbf{U}^{p=5\cdots200},$$

where $linespace(0, \frac{p}{10}, N_{\mathbf{X}})$ means samples $N_{\mathbf{X}}$ data from $[0, \frac{p}{10}]$.

**IHDP**  The generation process of $T$ and $Y$ are formulated as:

$$T = \frac{2\mathbf{U}^{p=1}}{5\mathbf{U}^{p=1}} + \frac{0.1 max(\mathbf{U}^{p=3,5,6}, 1)}{2.1 + min(\mathbf{U}^{p=3,5,6}, 1)} + \mathcal{N}(0, 0.25),$$

$$Y = \frac{sin(3T)\mathbf{U}^P T = 4}{1.2 - T} + \sum_{p=8}^{p=15} \mathbf{U}^p + \mathcal{N}(0, 0.25).$$

$\mathbf{U}$ are standardized to $\mathcal{N}(0, 1)$ and $T$ are normalized to $[0, 1]$.

The observed covariates $\mathbf{X}^{p=1\cdots25}$ are formulated as

$$\mathbf{X}^{p=1} = 0.2\frac{(\mathbf{U}^{p=1})^2}{\mathbf{U}^{p=2}}, \ \mathbf{X}^{p=2} = sin(\mathbf{U}^{p=3}),$$

$$\mathbf{X}^{p=3} = cos(\mathbf{U}^{p=4}), \ \mathbf{X}^{p=4} = 0.5exp(\mathbf{U}^{p=5}),$$

$$\mathbf{X}5 = \mathbf{U}^{p=6}\mathbf{U}^{p=7}, \ \mathbf{X}^{p=5\cdots25} = \mathbf{U}^{p=5}$$

**News**  The generation process of $\mathbf{T}$ and $Y$ are formulated as:

$$\mathbf{T}^{p=1\cdots5} = sin(\mathbf{U}^p)tanh(\mathbf{U}^p) + \mathcal{N}(0, 1),$$

$$Y^{p=1} = \frac{3\mathbf{T}^p(\mathbf{U}^p + 1) + \mathbb{M}(\mathbf{U}^{p=31\cdots100}, 1) + \mathcal{N}(0, 1)}{\epsilon}.$$

For News_2, $\epsilon = 2$, for News_5 $\epsilon = 4$. $\mathbf{U}$ are standardized to $\mathcal{N}(0, 1)$ and $\mathbf{T}$ are normalized to $[0, 1]$.

$$f_{\mathbf{X}}(\mathbf{U}^p) = \begin{cases} 0.2(\mathbf{U}^p)^2 & , \mathbb{I}\{mod((p-1), 5) \equiv 0\} \\ sin(\mathbf{U}^p) + 0.1 & , \mathbb{I}\{mod((p-1), 5) \equiv 1\} \\ cos(\mathbf{U}^p) + 0.1 & , \mathbb{I}\{mod((p-1), 5) \equiv 2\} \\ 0.1(10 + abs(\mathbf{U}^p) & , \mathbb{I}\{mod((p-1), 5) \equiv 3\} \\ abs(\mathbf{U}^p) & , \mathbb{I}\{mod((p-1), 5) \equiv 4\} \end{cases}$$

Given $f_{\mathbf{X}}$, the observed covariates $\mathbf{X}^{p=1\cdots2870}$ are formulated as

$$\mathbf{X}^{p=1\cdots20} = f_{\mathbf{X}}(\mathbf{U}^p),$$

$$\mathbf{X}^{p=21\cdots2870} = \mathbf{U}^p.$$

### A.4.2  DETAILS ON MODELS

We construct CRNet with depth 5. As Fig. 2 shown, $\mathcal{F}$ consists of 5 FCs with $\{256, 128, 128, 128, 128\}$ hidden units. $\mathcal{G}$ consists of 5 FCs with $\{32, 64, 64, 32, 32\}$ hidden units. The MDN module consists of 3 FCs with $\{20, 20, 20\}$.

NN consists of 4 FCs with $\{32, 32, 32, 1\}$ hidden units. We implement GPS, Bart, CF and DRNet following Schwab et al. (2020). We improve on DRNet and implement VCNet following Nie et al. (2021). We implement CEVAE following Louizos et al. (2017). We normalize simulation data to [0,1] for the conditional density estimator in DRNet and VCNet.

### A.4.3  DETAILS ON HYPERPARAMETERS

For GPS , Bart and CF, we use the default hyperparameters as Schwab et al. (2020). For all representation-based models, we fixed the random seed and search for the best performance with SGD or Adam. We also adjust the learning rate with $\{0.1, 0.01, 0.001, 0.0001, 0.00001\}$. For DRNet and VCNet, we adjust the hyperparameters knots with $\{[0.33, 0.66], [0.2, 0.4, 0.6, 0.8], [0.1, 0.2, 0.3, 0.4, 0.5, 0.6, 0.7, 0.8, 0.9]\}$ and $\alpha$ with $\{100, 10, 1, 0.1, 0.01, 0.001\}$. For VC-Net+TR, we adjust the hyperparameters $\beta$ with $\{10, 1, 0.1\}$ and the learning rate of TR with $\{0.1, 0.01, 0.001\}$. Besides, for CRNet, we adjust hyperparameters $\alpha$ with $\{100, 10, 1, 0.1, 0.01, 0.001\}$ and $\beta = 1$ consistently.

