# OpenReview forum: "Multi-Treatment Effect Estimation with Proxy: Contrastive Learning and Rank Weighting"
_ICLR.cc/2023/Conference — Submitted to ICLR 2023_

### Official Review · Reviewer_2J3F · 2022-10-24

**Confidence:** 3
**Correctness:** 3
**Technical Novelty And Significance:** 2
**Empirical Novelty And Significance:** 2
**Recommendation:** 3

**Clarity, Quality, Novelty And Reproducibility:**

I believe there are serious issues in the presentation that should be addressed. I didn't find this work to give a particularly novel insight into the problem in the context of related work.

**Strength And Weaknesses:**

**Strengths**
- The paper proposes new ideas to leverage the proxy model and show that the proposed approach outperforms in several example.
- The experiments consider many different baselines.

**Weaknesses**
- The paper has serious flaws in presentation that hinder the comprehension of the method, assumptions, and novelty. For instance, in the preliminaries section, a potential outcome is never defined, $\phi$ is not defined, the notation $\phi(\cdot | t)$ is not defined, Def. 2 and the notation do(t) independent of U are highly non standard and should be explained. Given this I don't understand the motivation section of the paper.
- Assumptions 5 and 6 should be discussed in more detail. It is not clear how plausible they are. Especially, assumption 6 seems to directly contradict the causal graph in Fig. 1.
- To my knowledge none of the semi-synthetic datasets incorporate a proxy structure in their data generating mechanisms. Are these changed to fit the setting in this paper. Do you loose performance when the correct thing to do is simply condition on observed covariates, i.e. if there are no unobserved confounders?

**Suggestions**
- One reference on ITE with continuous treatment could be added: Bellot, Alexis, Anish Dhir, and Giulia Prando. "Generalization bounds and algorithms for estimating conditional average treatment effect of dosage." arXiv preprint arXiv:2205.14692 (2022).

**Summary Of The Paper:**

Estimating treatment effect in the presence of unobserved confounders is a challenging but omnipresent problem. The authors assume availability of a proxy for unobserved confounders and attempt to estimate it to then infer weights that can be used for unbiased treatment effect estimation. The main contribution of the paper is the proposal of "rank weights" for this purpose.

**Summary Of The Review:**

My recommendation is to reject. I do believe that there are promising ideas and I am open to revise my score but at this moment there are many imprecisions that have to be corrected.

---

> ### Author Response · Authors · 2022-11-16
> **Response**
>
> **Q1. Assumptions 5 and 6 should be discussed in more
> detail. It is not clear how plausible they are. Especially, assumption 6 seems
> to directly contradict the causal graph in Fig. 1.**
>
> A1. Thanks for your concern. We make assumption 5
> following Louizos et al. (2017). Intuitively, it assumes that there are enough
> proxy variables in the data to enable us to capture the information of U from
> the observed data. Fig. 1 is the illustration of assumption 6.
>
> **Q2. To my knowledge none of the semi-synthetic datasets
> incorporate a proxy structure in their data generating mechanisms. Do you loose
> performance ... if there are no unobserved confounders?**
>
> A2. Thanks for your concern. As far as we know, the
> baseline Louizos et al. (2017) use the semi-synthetic datasets incorporate a
> proxy structure in their data generating mechanisms. And the performance with
> no unobserved confounders is in Table 2.
>
> [1] Louizos C, Shalit U, Mooij J M, et al. Causal
> effect inference with deep latent-variable models[J]. Advances in neural information
> processing systems, 2017, 30.

---

### Official Review · Reviewer_X9JX · 2022-10-24

**Confidence:** 4
**Correctness:** 2
**Technical Novelty And Significance:** 2
**Empirical Novelty And Significance:** 2
**Recommendation:** 5

**Clarity, Quality, Novelty And Reproducibility:**

Related work:
- The paper is missing relevant related works that also handle the case of hidden confounders for treatment effects estimation [1-3]

Reproducibility:
- The authors provide detailed descriptions of the datasets used for evaluation and the hyper parameters used by the model which I believe helps with reproduction.

Minor clarity issues.
- I believe that the clarity of the paper would improve if they authors mentioned what the expectations in Section 3.2 are computed over.
- The citation for Proxy (Guo et al., 2020) in Section 2 might be incorrect as the referenced paper is a survey paper.
- Typo “encode X representations”
- Typo in equation (10) where it should be $l_{Cr}$ instead of $L_{Er}$


[1] Wang, Yixin, and David Blei. "A proxy variable view of shared confounding." International Conference on Machine Learning. PMLR, 2021.
[2] Wang, Yixin, and David M. Blei. "The blessings of multiple causes." Journal of the American Statistical Association 114.528 (2019): 1574-1596.
[3] Mayer, Imke, et al. "Missdeepcausal: Causal inference from incomplete data using deep latent variable models." arXiv preprint arXiv:2002.10837 (2020).



**Strength And Weaknesses:**

Strengths:
-  The paper addresses an important and understudied problem in the causal inference literature, which is the estimation of treatment effects for continuous and multi-dimensional treatments in the challenging setting of unobserved confounders
- The use of contrastive learning to learn the proxy regularisation + the rank weighting to address the selection bias is interesting and aims to overcome the known problems of balanced representations which may result in too strong regularisation and discard information about the confounders
- CRNet is thoroughly evaluated on many datasets and demonstrates performance improvements over the benchmarks.


Weaknesses:
- The authors simplify the Average Dose-Response Function estimation problem through the additive regression model in equation (2). I think the paper would benefit from a more detailed discussion regarding how realistic this simplification is and in which cases it holds.
- The paper makes many assumptions; while I believe that this is needed to be able to handle the setting of unobserved confounders, the papers should include a more detailed discussion about the implications of these assumptions and how realistic they are in practice.
- It it unclear how learning the representation f(X) through contrastive learning helps with recovering the hidden confounders. In particular, why does strengthening the association between treatments T and covariates X helps to recover the unmeasured common cause U? I believe the authors should describe this in more detail, and if possible, develop theory to support this claim which I believe is crucial for the paper.
- To approximate the treatment score $\mathbb{P}(\mathbf{T} \mid \mathbf{U})$ the paper assumes it follows a Gaussian mixture model and uses a mixture density network to learn it. I believe that this assumption about the distribution of the treatments given the unobserved confounders should be more explicitly acknowledged and again discussed in which setting it is valid.



**Summary Of The Paper:**

The paper tackles the problem of estimating the effect of continuous and multi-dimensional treatments in the setting with unobserved confounders. The paper assumes access to high dimensional proxy variables for the unobserved confounders and proposes a Contrastive regulariser which learns a proxy representation for the unobserved confounders. The proposed method, CRNet, uses the Contrastive regulariser, together with a Rank weighting methods which de-biases the treatment assignment, to effectively estimate the effects of multiple continuous treatments.

**Summary Of The Review:**

Overall, I believe that the method developed in this paper, CRNet aims to address an important and understudied problem in the causal inference literature and achieves good performance in practice. However, I believe the authors should include more details regarding how the learnt representations recover the hidden confounders, as this is a crucial point for the paper. Moreover, the paper makes many explicit and implicit assumptions, whose implications are not discussed.

---

> ### Author Response · Authors · 2022-11-16
> **Response**
>
> Thank you for your careful reading and suggestions.
> Your suggestions are enlightening, and we have learned much from them. We
> sincerely look forward to your further suggestions.
>
> **Q1. ... the papers should include a more detailed
> discussion about the implications of these assumptions and how realistic they
> are in practice. ...**
>
> A1. This paper mainly focuses on the estimation
> problem. The identifiability analysis can be found in the appendix of Louizos
> et al., 2017. The interaction between T and X is an essential issue in the
> multiple continuous treatment effect estimation task. Existing estimation
> methods model T by explicit probability modeling (Louizos et al., 2017; Wu
> & Fukumizu, 2021) or discrete classification (Schwab et al., 2020; Nie et
> al., 2021; Ioana et al., 2020), and we believe that such a modeling approach
> cannot capture the distribution information of U better, so we propose to treat
> T and X as noise terms of U and perform instance discrimination by contrast
> learning, which can better capture the information of T and X. In this paper,
> we model T and X as noise terms of U for the causal scenario itself. We then
> consider T and X within the sample as positive samples of U. T and X after
> permutation as negative samples of U. By constraining the positive and negative
> samples, and we can maximize the information lower bound of U.
>
> **Q2.  ... missing relevant related works...**
>
> A2. The setting in (Yixin, 2021) is different from
> ours. Following the negative controls (Miao, 2018), (Yixin, 2021) assumes the
> treatments  must be
> non-empty. In our paper, we do not assume this setting. Furthermore, this paper
> does not focus on the identification task. What we want to convey is that the
> Cr proposed in this paper does not explicitly assume the probability model
> distribution of U in the treatment effect estimation as the probability model
> does, for instance, the substitute confounder in (Yixin, 2021).
>
> [1] Louizos C, Shalit U, Mooij J M, et al. Causal
> effect inference with deep latent-variable models[J]. Advances in neural
> information processing systems, 2017, 30.
>
> [2] Wang Y, Blei D. A proxy variable view of shared
> confounding[C]//International Conference on Machine Learning. PMLR, 2021:
> 10697-10707.
>
> [3] W. Miao, Z. Geng, and E. J. Tchetgen Tchetgen.
> “Identifying causal effects with proxy variables of an unmeasured confounder”.
> In: Biometrika 4 (2018).

---

> > ### Comment · Reviewer_X9JX · 2022-12-03
> > **Response**
> >
> > Thank you for your response and the clarifications! However, I still believe that the paper in its current format is not ready for publication. In particular, the paper still needs to discuss in more details the assumptions made and provide a more rigorous justification of the proposed method.

---

### Official Review · Reviewer_S7Ex · 2022-10-25

**Confidence:** 4
**Correctness:** 3
**Technical Novelty And Significance:** 3
**Empirical Novelty And Significance:** 2
**Recommendation:** 5

**Clarity, Quality, Novelty And Reproducibility:**

The perspective and particular way of using contrastive learning to achieve unbiased treatment effect estimation, to the best of my knowledge, carries novelty.

The readability of the paper is generally ok for readers with some background, and could be further improved. The argument of having the advantage of not requiring the prior is briefly mentioned in Section 1 and seems not much discussed after Section 1.

There are quite many typos and presentation issues in the paper. The following shows some spotted while I am reading the paper
p2 line 12: And with an **ideally** representation for confounders, we would adopt a balancing **methods**
p3 Definition 2: \Phi^* is not defined
p3 Section 3.2: "And the **the** estimated ..."
p3 Section 3.2: "Then we **the** divide ADRF estimation with "
p5 line 6: "these two responsibilities **complete** each other.
p5 line 6/7: " learned representation E(X) **refuse** the .."
p7 Eq 10: l_{Er} should be i_{Cr}

**Strength And Weaknesses:**

Strengths:
+ Unbiased continuous treatment effect estimation is an interesting and important problem.
+ The idea of using contrastive learning for learning the representation of observed proxy variables and the treatments provides an interesting perspective to address the problem as compared to the existing VAE and GAN approaches.

Weaknesses:
- The key contribution is the use of the Cr regularizer. The novelty related to the weighting for the score balancing is considered limited.
- Ablation study is missing. So, it is not clear about the relative importance of Cr and Rw.
- Only CEVAE was compared in the experiment while there are other more recent VAE models proposed for the same application (e.g., intact-VAE). Whether the proposed CRNet can perform the more recent extensions of CEVAE is unclear.
- The proposed approach may not be useful for applications where the cofounders are intended to be modeled explicitly.
- The evaluation is only carried out using synthetic and semi-synthetic data, and no real data is used.
- There are quite many typos and presentation issues in the paper as listed in the section of the review.


**Summary Of The Paper:**

This paper proposes the use of contrastive learning to deconfound the observed proxy variable of the confounder and treatment for unbiased continuous treatment effect estimation. In addition, some weighting methods are incorporated for score balancing. The proposed method has been evaluated using synthetic and sem-synthetic datasets. It shows that the use of contrastive learning approach can achieve the treatment effect estimation without the need to model the confounders expicitly.

**Summary Of The Review:**

The paper addressed an interesting and important problem. The perspective of using contrastive learning to solve the unbiased treatment effect estimation is an interesting one and carries some novelty. The performance comparison with more recent variants of CEVAE is not included. It is hard to evaluate the significance of this work.

---

> ### Author Response · Authors · 2022-11-16
> **Response**
>
> **Q1. The novelty related to the weighting for the score
> balancing is considered limited.**
>
> A1. Thanks for your concern. The Rw proposed in this
> paper is the first to model the weights as rank and stride and directly set
> stride to uniform distribution to obtain the rank weight. The experiments also
> demonstrate that the Rw is simple to implement and effective.
>
> **Q2. Ablation study is missing. So, it is not clear
> about the relative importance of Cr and Rw.**
>
> A2. The ablation experiments are presented in Table 2.
> The results demonstrate that Cr and Rw work well in treatment effect
> estimation. When we use it simultaneously, they will achieve significant
> improvement.
>
> **Q3. Only CEVAE was compared in the experiment while
> there are other more recent VAE models proposed for the same application (e.g.,
> intact-VAE). Whether the proposed CRNet can perform the more recent extensions
> of CEVAE is unclear.**
>
> A3. Thank you for your suggestion. But the intact-VAE
> focus on the binary treatment setting, and we do not find recent work similar
> to ours. There is no shortage of recent work in the articles we compared, which
> we believe is sufficient to demonstrate the performance of CRNet on the
> estimation task.
>
> **Q4. The proposed approach may not be useful for
> applications where the confounders are intended to be modeled explicitly.**
>
> A4. Motivated by contrastive learning, which uses a
> limited number of samples to capture high-quality representation by
> self-supervised learning, we propose to regularize the proxy representation
> with positive and negative pairs: given a batch of shuffled data S with n
> samples, the treatments and the covariates within the same samples are positive
> pairs; treatments and covariates across different samples are negative samples.
> We name it the Contrastive Regularizer (CR), which amplifies the correlation
> between the unit's treatment and covariates within-sample and weakens the
> anomalous correlations across-sample. Contrastive learning has been
> successfully applied in various fields thanks to its data-driven nature. Its
> lack of an explicit distribution prior makes it easier to use.
>
> **Q5. The evaluation is only carried out using synthetic
> and semi-synthetic data, and no real data is used.**
>
> A5. Thanks for your concern. As far as we know, no
> authoritative open-source data is available for continuous treatment effect
> estimation. Following (Louizos et al., 2017; Wu & Fukumizu, 2021; Schwab et
> al., 2020; Nie et al., 2021; Bica., 2020), we use synthetic and semi-synthetic
> data to evaluate our experiments.
>
> [1] Louizos C, Shalit U, Mooij J M, et al. Causal
> effect inference with deep latent-variable models[J]. Advances in neural
> information processing systems, 2017, 30.
>
> [2] Wu P, Fukumizu K. Intact-VAE: Estimating treatment
> effects under unobserved confounding[J]. arXiv preprint arXiv:2101.06662, 2021.
>
> [3] Schwab P, Linhardt L, Bauer S, et al. Learning
> counterfactual representations for estimating individual dose-response
> curves[C]//Proceedings of the AAAI Conference on Artificial Intelligence. 2020,
> 34(04): 5612-5619.
>
> [4] Nie L, Ye M, Liu Q, et al. Vcnet and functional
> targeted regularization for learning causal effects of continuous
> treatments[J]. arXiv preprint arXiv:2103.07861, 2021.
>
> [5] Bica I, Jordon J, van der Schaar M. Estimating the
> effects of continuous-valued interventions using generative adversarial
> networks[J]. Advances in Neural Information Processing Systems, 2020, 33:
> 16434-16445.

---

> > ### Comment · Reviewer_S7Ex · 2022-12-03
> > **Response**
> >
> > Thanks for the clarification, particularly for the ablation study and the dataset used for evaluation. Yet I still consider the novelty a concern, as echoed by the other reviewers.

---

### Official Review · Reviewer_kSiR · 2022-10-25

**Confidence:** 3
**Clarity, Quality, Novelty And Reproducibility:** See above.
**Correctness:** 3
**Technical Novelty And Significance:** 3
**Empirical Novelty And Significance:** 3
**Recommendation:** 5

**Strength And Weaknesses:**

**Strength**

- The proposed method is new to the best of my knowledge. Under the proxy assumption, the paper proposes to combine contrastive learning of the proxies and the rank weight estimations of the propensity scores to estimate treatment effects.
- The paper has relatively extensive set of baselines and plenty of tasks in the experimental analysis part. I also appreciate that the authors have detailed all synthetic functions in the appendix for reproducibility.

**Weakness**
- Missing important references and lacks clear motivation.  The problem setting of this paper is very closely related to (Yixin, 2021), which should definitely be mentioned and compared. I found it hard to be convinced by the author's argument that "existing methods rely on correctly specified treatment assignment models and/or require strong prior of the unobserved confounder distribution", since the identification results in (Wang, 2021) does not rely on any of these. Given those, the paper lacks key motivations for the Cr + Rw approach.

- The claim that "our method need no data distribution prior and outperforms others in performance" is not very convincing. To some degree, the choice of similarity functions in the contrastive learning part already implicitly assumes certain prior preferences. Even assuming it's true that this method requires absolutely no prior, this comes at a cost of: 1), one needs to specify the similarity function as well; 2), the identifiability of both treatment effects and latent confounders become unclear.

- Although the proposed method is new, I don't immediately see the key novelty of the paper. The contrastive learning is not a new method and has been used for learning latent representations for a long time; the rank weighting might have certain degree of novelty, however I don't see how this relates to the proxy causal learning setting. It looks more like a standalone estimator on this own and needs more theoretical analysis and experiment evaluation for justification.
- Lacks rigorous theoretical analysis. For example, the authors claim that " learned representation E(X) refuse the information of $\epsilon_1$ and maintain the information of $U$, which means we eliminate the error item...", which is too heuristic and needs to be formally derived. Also, the paper lacks identifiability analysis for $U$ (or treatment effects). Without those analysis, Eq(11) that the authors proposes hardly qualifies as an "identification"; and simply dodging this issue by hand-waving " The performance of using proxy to estimate ADRF depends on the degree of recovery of $U$" is clearly unconvincing.

**Minor issues**

- In related works, the paper cited a bunch of methodologies and claims that the proposed method "outperforms others in performance". Unfortunately, many of those mentioned in this part are not used in experiments at all. The authors need to be careful on what their claims and avoid over-selling.

- The writing of the paper is not very consistent. For some parts it is very clear and convincing, but for some other parts it is very confusing and I found it hard to understand what the authors are trying to do. I finally managed to understand most of it, but at the cost of reading it multiple times and making my own guess on a number of undefined terms and notations.


**Reference**

Wang, Yixin, and David Blei. "A proxy variable view of shared confounding." International Conference on Machine Learning. PMLR, 2021.

**Summary Of The Paper:**

This paper proposed a new model for causal proxy learning for average dose-response function estimation of multidimensional treatments.  This is done by based on combining contrastive regularizer to learn the proxy representation, and ranked weighting method to de-bias the treatment assignment mechanism estimation. The authors claims that the proposed method can avoid the deficiencies of existing methods that rely on correctly specified treatment assignment models and/or require strong prior of the unobserved confounder distribution.

**Summary Of The Review:**

Causal proxy learning is generally a very interesting topic, and this paper proposes a new solution to it. However, the motivation for the design choices of the paper is quite unclear; and the theoretical justification is not very convincing.

---

> ### Author Response · Authors · 2022-11-16
> **Response**
>
> Thank you for your careful reading and suggestions. We think your suggestions are very enlightening, and we have learned much from them. We give our views based on your suggestions and look forward to your further suggestions.
>
> **Q1. Missing important references and lacks clear motivation. The problem setting of this paper is very closely related to (Yixin, 2021), ... Given those, the paper lacks key motivations for the Cr + Rw approach.**
>
> A1. Thank you for your suggestion. The setting in (Yixin, 2021) is different from ours. Following the negative controls (Miao, 2018), (Yixin, 2021) assumes the treatments  must be non-empty. In our paper, we do not assume this setting. Furthermore, this paper does not focus on the identification task. What we want to convey is that the Cr proposed in this paper does not explicitly assume the probability model distribution of U in the treatment effect estimation as the probability model does, for instance, the substitute confounder in (Yixin, 2021).
>
> **Q2.The claim that "our method need no data distribution prior and outperforms others in performance" is not very convincing.**
>
> A2. Thanks for your concern. We make substantially
> weaker assumptions in Section 3.3 about the data-generating process and the
> structure of the hidden confounders, i.e., negative control setting or proxy
> variable setting.
>
> **Q3. Although the proposed method is new, I don't
> immediately see the key novelty of the paper. The contrastive learning is not a
> new method …;**
>
> A3. Thank you for your concern. The novelty of Cr is
> divided into two points. One point is that we introduce contrastive learning
> into the causal effect estimation task. In the multiple continuous treatment
> effect estimation task, the interaction between T and X is an important issue.
> Existing estimation methods model T by explicit probability modeling (Louizos
> et al., 2017; Wu & Fukumizu, 2021) or discrete classification (Schwab et
> al., 2020; Nie et al., 2021; Ioana et al., 2020), and we believe that such a modeling
> approach cannot capture the distribution information of U better, so we propose
> to treat T and X as noise terms of U and perform instance discrimination by
> contrast learning, which can better capture the information of T and X. In this
> paper, we model T and X as noise terms of U for the causal scenario itself. We
> then consider T and X within the sample as positive samples of U and T and X
> after permutation as negative samples of U. By constraining the positive and
> negative samples, and we can maximize the information lower bound of U.
>
> **Q4. the rank weighting might have certain degree of
> novelty, however I don't see how this relates to the proxy causal learning
> setting. It looks more like a standalone estimator on this own ...**
>
> A4. Thank you for your interest. Indeed, in this
> paper, we have two separate contributions, CR block for representation and Rank
> weight for weighting, both of which serve the setting of multiple continuous
> treatments and perform better when used in combination, and the ablation experiments
> are shown in Table 2.
>
> **Q5. ... the paper lacks identifiability analysis
> for …**
>
> A5. Thanks for your concern. This paper mainly focuses
> on the estimation problem. The identifiability analysis can be found in the
> appendix of Louizos et al., 2017.
>
> **Q6. Many of those mentioned in this part are not used
> in experiments at all. The authors need to be careful on what their claims and
> avoid over-selling.**
>
> A6. Thanks for your concern. In the related work, we
> intend to express how this paper differs from others, many of which (e.g.,
> intact-vae, negative control) are related to us but different from our setup,
> and in work with a similar setup to ours (e.g., DRNet, VCNet, CEVAE), our
> proposed approach works better than theirs. In order to avoid ambiguity, we
> highlight some work similar to our work in the final paper.

---

> > ### Author Response · Authors · 2022-11-16
> > **Reference of Response**
> >
> > [1] Wang Y, Blei D. A proxy variable view of shared
> > confounding[C]//International Conference on Machine Learning. PMLR, 2021:
> > 10697-10707.
> >
> > [2] W. Miao, Z. Geng, and E. J. Tchetgen Tchetgen.
> > “Identifying causal effects with proxy variables of an unmeasured confounder”.
> > In: Biometrika 4 (2018).
> >
> > [3] Louizos C, Shalit U, Mooij J M, et al. Causal
> > effect inference with deep latent-variable models[J]. Advances in neural
> > information processing systems, 2017, 30.
> >
> > [4] Wu P, Fukumizu K. Intact-VAE: Estimating treatment
> > effects under unobserved confounding[J]. arXiv preprint arXiv:2101.06662, 2021.
> >
> > [5] Schwab P, Linhardt L, Bauer S, et al. Learning
> > counterfactual representations for estimating individual dose-response curves[C]//Proceedings
> > of the AAAI Conference on Artificial Intelligence. 2020, 34(04): 5612-5619.
> >
> > [6] Nie L, Ye M, Liu Q, et al. Vcnet and functional
> > targeted regularization for learning causal effects of continuous
> > treatments[J]. arXiv preprint arXiv:2103.07861, 2021.
> >
> > [7] Bica I, Jordon J, van der Schaar M. Estimating the
> > effects of continuous-valued interventions using generative adversarial
> > networks[J]. Advances in Neural Information Processing Systems, 2020, 33:
> > 16434-16445.
> >
> > [8] Huang W, Yi M, Zhao X. Towards the generalization
> > of contrastive self-supervised learning[J]. arXiv preprint arXiv:2111.00743,
> > 2021.
> >
> > [9] Johansson F D, Shalit U, Kallus N, et al.
> > Generalization bounds and representation learning for estimation of potential
> > outcomes and causal effects[J]. arXiv preprint arXiv:2001.07426, 2020.

---

> > ### Comment · Reviewer_kSiR · 2022-12-02
> > **Response**
> >
> > Thank you for the clarification. It has address some of my concerns and I have increased my score. However, I still share some concerns with other reviewers regrading theoretical justification (on assumptions and identifiability theory) and empirical results.

---

### Decision · Program_Chairs · 2023-01-20

**Decision:**

Reject

**Justification For Why Not Higher Score:**

The paper still needs to discuss in more detail the assumptions made and provide a more rigorous justification of the proposed method.


**Justification For Why Not Lower Score:**

N/A

**Metareview: Summary, Strengths And Weaknesses:**

This paper proposes the use of contrastive learning to deconfound the observed proxy variable of the effect estimation.

strength

+ The paper proposes new ideas to leverage the proxy model and show that the proposed approach outperforms in several examples.

+ The use of contrastive learning to learn the proxy regularisation + the rank weighting is interesting. It could potentially overcome the known problems of balanced representations which may result in too strong regularisation and discard information about the confounders

weakness

+ The paper needs to discuss in more detail the assumptions, esp assumptions 5 and 6.

+ The paper needs to provide a more rigorous justification of the proposed method (esp on the identifiability theory) and more thorough empirical validation. The evaluation is only carried out using synthetic and semi-synthetic data, and no real data is used.

+ The current framing of the paper does not convey clear motivation nor reveal key novelty of the paper.

The paper is not ready for publication in its current form. We encourage the authors to make revisions and submit to a future venue.


**Summary Of Ac-Reviewer Meeting:**

N/A